# Dual Inhibition of the Renin–Angiotensin–Aldosterone System and Sodium–Glucose Cotransporter-2: Mechanistic and Clinical Evidence for Cardiorenal Protection

**DOI:** 10.3390/biomedicines14010101

**Published:** 2026-01-03

**Authors:** Reem F. M. Aazar, Rayan Arzouni, Persoulla A. Nicolaou

**Affiliations:** Department of Basic and Clinical Sciences, University of Nicosia Medical School, Nicosia 2417, Cyprus; aazar.r@live.unic.ac.cy (R.F.M.A.); arzouni.r@live.unic.ac.cy (R.A.)

**Keywords:** renin–angiotensin–aldosterone system (RAAS), sodium glucose co-transporter inhibitors (SGLT2i), heart failure (HF), type 2 diabetes (T2D), angiotensin converting enzyme inhibitor (ACEI), angiotensin receptor blocker (ARB), angiotensin receptor-neprilysin inhibitor (ARNI), mineralocorticoid receptor antagonist (MRA), cardiorenal protection

## Abstract

Overactivation of the renin–angiotensin–aldosterone system (RAAS) promotes haemodynamic overload, inflammation, and fibrosis in the heart and kidneys. Recently, sodium–glucose cotransporter-2 (SGLT2) inhibitors have emerged as a cornerstone therapy in cardiorenal protection. Emerging data indicate that adding SGLT2 inhibitors to angiotensin-converting enzyme (ACE) inhibitors, angiotensin receptor blockers, mineralocorticoid receptor antagonists, or angiotensin receptor–neprilysin inhibitors confers additional cardiorenal protection, yet their mechanistic basis and optimal clinical use in cardiovascular (CV) disease remain unclear. This review will integrate pre-clinical and clinical evidence on dual RAAS/SGLT2 modulation in CV disease, providing mechanistic insight into dual therapy. The review will finally outline priorities for future translational and outcome studies. Clinically, adding SGLT2 inhibitors to RAAS-based therapy reduces heart failure hospitalizations and slows kidney disease progression without new safety liabilities in type 2 diabetes, heart failure, and chronic kidney disease. Mechanistically, SGLT2 inhibition restores tubuloglomerular feedback and constricts the afferent arteriole; RAAS blockade dilates the efferent arteriole, and together, they lower intraglomerular pressure. Both classes also reduce oxidative stress, inflammatory signalling, and pro-fibrotic pathways, with SGLT2 inhibitors in several settings shifting RAAS balance toward the protective ACE2/angiotensin-(1–7)/Mas receptor axis. Key gaps include the scarcity of adequately powered trials designed to test combination therapy versus either component alone, limited evidence on timing and sequencing, incomplete characterization in high-risk groups, and mechanistic insight limited by study design in animal and cell models. Collectively, current data support layering SGLT2 inhibitors onto RAAS-based therapy, while definitive evidence from dedicated clinical trials is awaited.

## 1. Introduction

The renin–angiotensin–aldosterone system (RAAS) is a central regulator of cardiovascular and renal physiology, maintaining blood pressure (BP), electrolyte balance, and vascular tone. Activation of the system through its key effectors, renin, angiotensin II (Ang II), and aldosterone, represents an important homeostatic mechanism in states of reduced BP or renal perfusion [1,2]. While essential for homeostasis, sustained or dysregulated RAAS activity raises systemic and intraglomerular pressure, promotes sodium retention and neurohormonal stress, and accelerates tissue injury through oxidative stress, inflammation, and fibrosis in the heart and kidneys, contributing to the pathogenesis of CV and renal diseases, including primary hypertension, heart failure (HF), and chronic kidney disease (CKD) [2]. Consequently, over the past three decades, pharmacological RAAS blockade with angiotensin-converting enzyme inhibitors (ACEIs), angiotensin receptor blockers (ARBs), mineralocorticoid receptor antagonists (MRAs), and, more recently, angiotensin receptor–neprilysin inhibitors (ARNIs) has become a cornerstone of therapy. RAAS blockade mediates dilation of resistance vessels and reduces afterload, decreasing preload through natriuresis and venodilation, as well as efferent arteriole dilation, which reduces intraglomerular pressure and anti-fibrotic signalling. These pharmacological effects translate to reduced mortality, hospitalization, and disease progression across hypertension, CKD, and HF [2]. Clinical benefits need to be balanced against adverse effects, which include, amongst others, hyperkalaemia, hypotension, kidney injury, cough and angioedema with ACEIs [1,2]. More recently, sodium–glucose cotransporter 2 inhibitors (SGLT2i) have emerged as an important therapeutic class in cardiorenal protection. Initially developed as glucose-lowering agents for the management of type 2 diabetes mellitus (T2D), SGLT2 inhibitors confer multiple benefits beyond glycaemic control [3]. SGLT2 inhibitors block sodium–glucose cotransporter 2 in the proximal tubule, lowering glucose and sodium reabsorption, increasing distal NaCl delivery, and restoring tubuloglomerular feedback. The afferent arteriole constricts and intraglomerular pressure falls, along with anti-inflammatory and anti-fibrotic effects in the kidney, vasculature, and myocardium. Common drug-related risks include genital mycotic infections, volume depletion or hypotension, rare euglycaemic ketoacidosis in susceptible states, very rare necrotizing fasciitis of the perineum, amputation, and fracture [3]. These pharmacological actions improve cardiorenal outcomes in patients with or without T2D and have positioned SGLT2 inhibitors as essential components of guideline-directed medical therapy (GDMT) in T2D, CKD, and HF, in addition to standard therapy that includes RAAS blockade [3]. A growing body of pre-clinical and clinical work indicates that RAAS and SGLT2 inhibition act through complementary pathways and crosstalk with each other. Given these potential synergistic effects, dual inhibition of these pathways, with dual, triple, or quadruple therapy, has become a subject of significant interest. However, many questions remain about the clinical benefits, risks, and mechanistic basis of dual therapy. This review synthesizes current evidence on dual RAAS and SGLT2 inhibition with a focus on cardiorenal disease, including HF and T2D with CV disease. It critically evaluates clinical data and mechanistic insights and identifies areas where evidence remains limited.

We conducted a targeted literature search to ensure comprehensive evaluation of the latest research findings. We searched MEDLINE/PubMed, Scopus, and ScienceDirect up to 30 September 2025. Search terms included combinations of ‘Sodium glucose cotransporter inhibitor’, ‘SGLT2i’, ‘SGLT2 inhibitor’, ‘SGLT inhibitor’, ‘gliflozin’, ‘RAAS’, ‘Renin’, ‘Angiotensin’, ‘Aldosterone’, ‘ACEI’, ‘ARB’, ‘ARNI’, ‘Cardiovascular’, ‘Heart failure’, ‘Hypertension’, ‘Clinical trials’, ‘Mechanism’. Boolean operators ‘AND’ and ‘OR’ were used to refine results. Review articles and studies involving patient populations without CV disease or risk factors were excluded. The literature search yielded 64 clinical studies and 56 mechanistic studies. The abstracts were screened and full texts were retrieved and critically appraised for those studies aligning with the research question. We also screened the reference lists of all included studies and retrieved full texts for items meeting our review objectives.

## 2. Clinical Trials

The available clinical evidence for dual therapy comes largely from trials in which a SGLT2i was added to background therapies, with effects examined in pre-specified or post hoc sub-groups. Studies explicitly designed to test the combination against either component alone are scarce. Considering the initial clinical use of SGLT2i, evidence about CV benefits was first reported in T2D patients with established CV disease or risk factors for it. We firstly discuss the data below from T2D patients and then for patients with CV disease, primarily HF, without T2D. The findings are summarized in Table 1.

### 2.1. T2D and CV Risk Factors

Initial randomized trials evaluated SGLT2i on a universal RAAS inhibitor (RAASi) background. In Weber et al. [4], 449 patients with T2D and hypertension receiving an ACEI or ARB were randomized to dapagliflozin or placebo. Dapagliflozin reduced systolic BP versus placebo, (placebo-adjusted, −4.28 mmHg; 95% CI −6.54 to −2.02, *p* = 0.0002) and improved glycaemic control, with reduction in glycated hemoglobin (HbA1c) [4]. Similarly, in the SACRA trial [5], which enrolled 132 patients with T2D and inadequately controlled nocturnal hypertension, all patients were receiving an ARB. Empagliflozin reduced night-time systolic BP more than placebo (−6.3 vs. −2.0 mmHg), lowered HbA1c and uric acid, and produced an early non-progressive estimated glomerular filtration rate (eGFR) dip (69.8 vs. 65.0 mL/min/1.73 m^2^); atrial natriuretic peptide (ANP) and urine albumin-to-creatinine ratio (UACR) also fell. Both studies showed favourable tolerability, with mostly mild events such as thirst, polyuria, genital itching [5], and renal and volume depletion events, which were similar between groups [4]. There were no significant adverse effects that warranted the termination of combination therapy [4]. These early trials indicate that adding an SGLT2i, alongside RAAS blockade, improves BP and metabolic markers without new safety concerns.

Most subsequent information in diabetes with CV complications comes from pre-specified sub-group or post hoc analyses within larger programmes, as well as observational cohorts. In three large T2D populations with established symptomatic atherosclerotic cardiovascular disease (ASCVD) or multiple CV risk factors, SGLT2 inhibitors improved CV outcomes without evidence of a treatment by pre-specified sub-group interaction by ACEI, ARB, or MRA use [6,7,8]. Specifically, canagliflozin, (CANVAS, n = 10,142), empagliflozin, (EMPA-REG OUTCOME, n = 7020), and dapagliflozin (DECLARE–TIMI 58, n = 17,160) reduced the composite of CV death and HF hospitalization by 17–34% (HR 0.76–078), with benefits consistent among patients receiving ACEIs/ARBs [6,7,8]. The effects of ARNIs were not investigated, which is a limitation of the studies. Known class and drug specific safety signals for SGLT2i included higher risks of amputation and fractures and more volume depletion events, while osmotic diuresis-related events were fewer in patients with baseline HF [6]. The effect of combining SGLT2i with RAASi on adverse effects was not investigated in CANVAS [6] or EMPA-REG OUTCOME [7]. DECLARE-TIMI showed no excess adverse events across medication strata and a non-significant trend toward fewer hyperkalaemia events with MRA [8].

In analyses focused on kidney and BP outcomes in T2D patients with CV disease, SGLT2 inhibition, on top of RAAS blockade, consistently improved renal markers and lowered BP. A pooled post hoc analysis by Heerspink et al. [9], in patients with T2D, hypertension, and albuminuria, found that adjunctive dapagliflozin, when added to existing ACEI or ARB, reduced albuminuria by 33.2% and improved systolic BP and HbA1c, although the analysis did not isolate SGLT2i effects or adverse effects of a RAASi background [9]. In the EMPA-REG BP trial (n = 824), post hoc analysis showed that empagliflozin lowered systolic and diastolic BP similarly regardless of the number of background antihypertensives and whether ACEI or ARB was used, in patients with T2D and hypertension [10]. Genital mycotic infections were more frequent with empagliflozin, volume depletion events were rare, and orthostatic test positivity increased without excess hypotension-related adverse events. Overall drug-related adverse effects were similar across groups, independent of the number of background drugs [10]. Post hoc analysis from EMPA-REG OUTCOME (n = 7020), in patients with T2D and established CV disease, showed that empagliflozin reduced incident or worsened nephropathy (12.7% vs. 16.8%) across sub-groups, including ACEI or ARB users, and slowed progression to macroalbuminuria, with a borderline stronger effect in ACEI or ARB users [11]. The early eGFR dip (3–5 mL/min/1.73 m^2^) was modestly larger in ACEI or ARB users by less than 1 mL/min/1.73 m^2^ on average but without long-term consequence, and there was no increase in serious adverse events or discontinuation with the combination, further supporting the cardiorenal protection of combination treatment [11].

Observational studies corroborate the CV benefits of combining SGLT2 inhibition with RAAS blockade. In a retrospective, observational cohort study in patients with T2D and HF with reduced ejection fraction (HFrEF), Kim HM and others [12] investigated the effect of receiving SGLT2 + ARNI, ARNI alone, SGLT2 alone, or neither. Patients on dual therapy had fewer HF hospitalizations and CV deaths (HR 0.18, *p* = 0.006) and greater echocardiographic improvements, compared with other regimens. While the small number of participants (n = 206) is a limitation of the study, it provided further insight into the synergistic effects of these drugs [12]. The Korean National Health Database Cohort [13] carried out a similar observational cohort study, in a large population of 261, 783 patients in patients with T2D and hypertension, and showed that SGLT2i plus RAASi was associated with the lowest risk of a composite of end-stage kidney disease or death (HR: 0.68) and all-cause death (HR: 0.68), with larger relative benefits among older or higher-risk patients. As observational designs, these studies are subject to confounding and often lack complete safety ascertainment, but they are directionally consistent with clinical trial data [13].

Two small trials were designed specifically to test the effects of dual therapy. Karalliedde et al. [14] randomized 33 patients with T2D and persistent microalbuminuria to ramipril plus dapagliflozin or ramipril alone, and found that the combination reduced urinary albumin excretion by about 44%, while ramipril alone had no significant effects. Lastly, Lytvyn et al. [15] conducted a randomized double-blind crossover trial in 30 adults with T1D and renal hyperfiltration and showed that empagliflozin plus ramipril reduced GFR by 8 mL/min/1.73 m^2^ versus ramipril alone, and further lowered mean arterial pressure, systolic and diastolic BP, and total peripheral resistance. Adverse event profiles were similar across the two groups [14]; ketosis occurred without diabetic ketoacidosis, and genitourinary infections, and hypotension were infrequent and manageable [15]. These small studies support complementary glomerular and haemodynamic effects with the combination, although they are underpowered for clinical outcomes [15].

### 2.2. Heart Failure

Large randomized controlled trials in HF tested SGLT2i on contemporary GDMT and reported consistency across RAASi drugs in pre-specified sub-groups. DAPA-HF [16] enrolled 4744 patients with HFrEF and showed that dapagliflozin reduced the primary composite outcome of CV or HF deaths, wherein 16.3% vs. 21.2%, HR 0.74 [16]. Benefits were consistent across baseline ARNI and MRA use. The EMPEROR-Reduced study [17] in HFrEF patients similarly demonstrated a reduction in CV death or first HF hospitalization from 24.7% to 19.4% and a slower eGFR decline by 1.73 mL/min/1.73 m^2^ per year, with a renal composite HR of 0.50, when using another SGLT2i, empagliflozin, with no heterogeneity by ARNI use on formal interaction testing [17]. In HF and preserved ejection fraction (HFpEF), the beneficial effects of SGLT2i and RAAS blockade were evident in EMPEROR-Preserved (n = 5988) [18], which found a reduction in the primary composite (CV death or first HF hospitalization) from 17.1% to 13.8%, HR 0.79, with effects consistent across ACEI, ARB, ARNI, and MRA users. Genital or urinary infections and hypotension were more frequent with empagliflozin, although no sub-group analysis was conducted [18]. In post-myocardial infarction (MI), for patients at risk of HF or congestion (EMPACT-MI, n = 6522) [19], empagliflozin, in addition to GDMT, did not reduce the primary outcome (first HF hospitalization or all-cause death), but it reduced first HF hospitalization (HR 0.77, 95% CI 0.60–0.98, *p* = 0.031) and total HF hospitalizations (rate ratio 0.67, 95% CI 0.51–0.89, *p* = 0.006), with consistent effects across RAASi sub-groups and less need to initiate diuretics, ARNI, ACEI, or ARB, and MRA [19]. In HF with mildly to moderately reduced EF and chronic functional mitral regurgitation (MR), the smaller EFFORT trial (n = 128) showed improved MR severity and reverse remodelling with ertugliflozin, which is consistent within the ARNI sub-group [20]. Absolute adverse event counts were low and sub-group safety analyses were not presented [20]. These trials establish that SGLT2i provides cardiac and renal benefits across a range of phenotypes on top of RAASi-based regimens, without evidence of negative safety interactions, although none was designed to isolate a treatment interaction between RAASi and SGLT2i.

Several analyses focused specifically on background therapy post hoc sub-group analysis. A therapy interaction analysis of DAPA-HF [21] by Docherty et al. showed that dapagliflozin reduced the primary composite across drug and device sub-groups, including MRA and ARNI, with hazard ratios ranging from 0.57 to 0.86 and no treatment by sub-group interaction. Consistently, in further post hoc analyses from DAPA-HF, CV and renal benefits were similar with or without ARNI [22] or MRA [23]. Volume depletion events were slightly more frequent with loop diuretics or higher dose of MRA [21], while renal adverse events were not increased [21,22,23]. Among MRA users, dapagliflozin reduced moderate or severe hyperkalaemia (1.3% vs. 2.4%, HR 0.50) [23]. In HF with improved EF (HFimpEF), (DELIVER, n = 1.151), addition of dapagliflozin to GDMT reduced worsening HF or CV death across sub-groups defined by 0–1, 2, or 3 background HF therapy classes (ACEI or ARB or ARNI, β-blocker, and MRA). Nonetheless, the largest relative reduction was observed in those on 0–1 classes (HR 0.46). Importantly SGLT2i therapy did not necessitate the reduction in existing therapy doses, and adverse events were not increased across groups [24]. Finally, the FINEARTS-HF trial [25] randomized 6001 patients with HF with mildly reduced or preserved ejection fraction to finerenone (MRA) versus placebo. Finerenone reduced CV death and total worsening HF events (rate ratio 0.84) with similar relative benefits, whether or not patients were on an SGLT2i. Hyperkalaemia was more frequent with finerenone regardless of SGLT2i, and the early eGFR dip (−2.72 mL/min/1.73 m^2^, finerenone vs. −0.28 mL/min/1.73 m^2^, placebo) was similar with or without SGLT2i. These data inform the compatibility and safety of combining an MRA with an SGLT2i [25].

Complementary evidence comes from computational modelling that combined results from three trials (EMPHASIS-HF, PARADIGM-HF, and DAPA-HF) to decipher the effects of comprehensive quadruple therapy (ARNI + β-blocker + MRA + SGLT2i) vs. conventional therapy (ACEI/ARB + β-blocker) in patients with HFrEF. Quadruple therapy reduced CV death and first HF hospitalization by a significant 62%, as compared to conventional therapy, with sizeable gains in event-free (2.7–8.3 years) and overall survival (1.4–6.3 extra years of life). Although these findings are interesting, these projections are generated by hypothesis rather than confirmatory [26]. More recently, two studies were specifically designed to study the effects of dual therapy. In a single-centre observational cohort of HFrEF patients [27], adding an SGLT2i to optimized ARNI was associated with improved ventricular and atrial volumes, better systolic function, and lower N-terminal pro–B-type natriuretic peptide (NT proBNP) at 3 and 12 months [27]. Finally, the NovCon Sequencing Study [28] is a randomized protocol comparing rapid 4-week sequencing of β-blocker, SGLT2i, ARNI, and MRA with conventional stepwise initiation over about 6 months. Results for the primary composite of all-cause death and heart failure hospitalization and key secondary outcomes are pending [28].

Beyond the cardiorenal endpoints emphasized here, SGLT2i may also confer benefits not systematically assessed in these combination-therapy studies, for example, reductions in arrhythmic events. A 2025 meta-analysis of eight adjudicated randomized trials in T2D, HF, or CKD (n = 58,569) found that SGLT2i reduced sudden cardiac death, with an odds ratio of 0.82 (95% CI 0.72–0.94; *p* = 0.0104), suggesting beneficial anti-arrhythmic effects [29]. The study did not analyze effect modification by background RAASi use.

In summary, across diabetes and HF, SGLT2i consistently delivers cardiorenal benefits when added to ACEI, ARB, ARNI, or MRA, and combination use has not revealed new significant safety liabilities. The evidence base, however, is dominated by sub-group and post hoc analyses within larger trials and by observational cohorts. There is a scarcity of clinical trials to assess the effectiveness and safety of combination therapy.

**Table 1 biomedicines-14-00101-t001:** Dual RAAS and SGLT inhibition in cardiovascular disease: evidence from clinical trials.

Type 2 Diabetes and Cardiovascular Disease							
Study No.	Drugs	Study Name, Reference	Patient No.	Patient Population	Study Design	Clinical Outcomes	Adverse Effects
1	Dapagliflozin vs. Placebo (in addition to GDMT)	Weber et al., 2016 [4]	449	T2D and hypertension	Multicentre, prospective, randomized, placebo-controlled, double-blind clinical trial	Greater SBP reduction vs. placebo, placebo-adjusted difference −4.28 mmHg (95% CI −6.54 to −2.02, *p* = 0.0002) Improved glycaemic control by an additional HbA1c −0.61% (95% CI −0.76 to −0.46, *p* < 0.0001). All patients received ACEI or ARB	AEs, including renal and volume-depletion events, were infrequent and similar between groups
2	Empagliflozin vs. Placebo (in addition to ARB)	SACRA, Kario et al., 2019 [5]	132	T2D and nocturnal hypertension	Multicentre, prospective, randomized, placebo-controlled, double-blind clinical trial	Night-time SBP reduced with empagliflozin (−6.3 mmHg) vs. placebo (−2.0 mmHg) Reductions in HbA1c, uric acid and an early no-progressive eGFR dip observed with empagliflozin Reductions in ANP and UACR with empagliflozin	Combination therapy was well-tolerated; mild AEs included thirst, polyuria, genital itching, constipation
3	Canagliflozin vs. Placebo (in addition to GDMT)	CANVAS programme, Rådholm et al., 2018 [6]	10,142	T2D and symptomatic ASCVD or increased CV risk	Multicentre, prospective, randomized, placebo-controlled, double-blind clinical trial, with pre-specified sub-group analysis	Lower risk of CV death or hospitalized HF events vs. placebo, HR 0.78 (95% CI 0.67–0.91) Effect greater in those with HF at baseline Results were comparable across sub-groups, including RAASi users	Increased risk of amputation, fractures, and volume depletion osmotic Diuresis-related events were lower in HF patients Medication usage sub-groups not analyzed
4	Empagliflozin vs. Placebo, analyzed by baseline attainment of CV risk factor goals (one goal was ACEI/ARB therapy)	EMPAREG OUTCOME risk-factor-control analysis, Inzucchi et al., 2020 [7]	7020	T2D and established ASCVD	Post hoc sub-group analysis of a multicentre, randomized, placebo-controlled, double-blind trial	Empagliflozin reduced CV death (−38%), HF hospitalization (−35%), CV death + HF hospitalization, and three-point MACE uniformly across goal categories No treatment-by-sub-group interaction, including ACEI/ARB use	Safety not reported beyond the parent study
5	Dapagliflozin vs. Placebo (in addition to GDMT), analyzed by baseline use of ACEIs/ARBs, beta-blockers, diuretics, MRAs	DECLARE-TIMI 58, Oyama et al., 2022 [8]	17,160	T2D and established ASCVD or multiple CV risk factors	Multicentre, prospective, randomized, placebo-controlled, double-blind clinical trial, with pre-specified sub-group analysis	Dapagliflozin reduced CV death or HF hospitalization and a kidney composite irrespective of background ACEI/ARB, β-blocker, diuretic, or MRA Among triple therapy users (ACEI/ARB, beta-blocker, diuretic), dapagliflozin reduced CV death/HF hospitalization HR 0.76 (95% CI 0.62–0.93) and the kidney composite HR 0.62 (95% CI 0.44–0.87)	No AE increase with dapagliflozin across CV medications including RAASi Trend toward fewer hyperkalemia events with MRA (interaction *p* = 0.10, not significant)
6	Dapagliflozin + ACEIs or ARBs vs. Placebo + ACEIs or ARBs	Heerspink et al., 2016 [9]	356	T2D, hypertension and albuminuria	Post hoc analysis of a subset of patients from two randomized, placebo-controlled, double-blind clinical trials	Adjunctive dapagliflozin reduced albuminuria by 33.2% (95% CI 45.4, 18.2) vs. placebo SBP and HbA1c improved The effect of SGLT2i without RAASi background was not elucidated	Reversible eGFR reduction in SGLT2i group Safety similar to placebo and acceptable; serious events were infrequent, including volume depletion and UTIs (not significant) AEs not studied without RAASi background
7	Empagliflozin vs. Placebo (in addition to GDMT), analyzed by baseline use of antihypertensives and diuretic or ACEI/ARB	EMPAREG BP TRIAL, Mancia et al., 2016 [10]	824	T2D and hypertension	Post hoc sub-group analysis from a multicentre, prospective, randomized, placebo-controlled, double-blind clinical trial	SBP/DBP declined similarly across 0, 1, ≥2 antihypertensives (Pinteraction SBP 0.448, DBP 0.498) Effects were consistent with or without ACEI/ARB (Pinteraction SBP 0.900, DBP 0.359)	Overall and drug-related AEs similar across strata independent of number of background drugs Volume-depletion events rare; genital mycotic infections increased with empagliflozin; orthostatic test positivity higher but without hypotension-related AEs
8	Empagliflozin vs. Placebo (in addition to GDMT), analyzed by baseline drugs, including ACEI/ARB	EMPAREG OUTCOME TRIAL, Mayer et al., 2019 [11]	7020	T2D with established CV disease	Post hoc sub-group analysis from a multicentre, prospective, randomized, placebo-controlled, double-blind clinical trial	Lower risk of incident/worsening nephropathy across strata including ACEI/ARB Renal composite benefits were consistent across ACEI/ARB users Progression to macroalbuminuria reduced overall, borderline stronger with ACEI/ARB (interaction *p* = 0.045)	SGLT2i + ACEI/ARB did not increase serious AEs or discontinuation Early eGFR dip modestly larger in ACEI/ARB users but without long-term consequence Genital infections more frequent with empagliflozin
9	Group 1: ARNI + SGLT2i; Group 2: ARNI only; Group 3: SGLT2i only; Group 4: neither	Kim et al., 2021 [12]	206	T2D and HFrEF	Multicentre, retrospective, observational cohort study	Combination therapy lowered HF hospitalization and CV mortality vs. others (HR 0.18, *p* = 0.006); greater echo improvements and EF gains Larger improvements when initiating ARNI vs. SGLT2i	Not reported
10	Group 1: No RAASi or SGLT2i Group 2: SGLT2i only Group 3: RAASi only 4: SGLT2i + RAASi	Korean National Health Database Cohort, Hong et al., 2025 [13]	261,783	T2D and hypertension	Retrospective observational cohort study	Combination SGLT2i + RAASi showed greatest protection: composite ESKD/death HR 0.68; all-cause death HR 0.68; ESKD HR 0.63 (not significant). Benefits greatest in higher-risk sub-groups (age over 65, longer diabetes/hypertension, proteinuria).	Safety outcomes not reported
11	Ramipril vs. Ramipril + Dapagliflozin	Karalliedde et al., Front 2022 [14]	33	T2D with persistent microalbuminuria	Single-centre, prospective, parallel randomized clinical trial	Combination reduced urinary albumin-excretion rate by ≈44%; ramipril alone had no significant change. No change in arterial stiffness/ageing	Overall adverse effect profile was similar between the groups
12	Empagliflozin + Ramipril vs. Placebo + Ramipril	Lytvyn et al., 2022 [15]	30	T1D and renal hyperfiltration	Single-centre, prospective, randomized, placebo-controlled, double-blind crossover	Combination decreased GFR vs. ramipril alone by 8 mL/min/1.73 m^2^ Greater proximal sodium and fluid reabsorption Additional reductions in MAP, SBP (−4 mmHg), DBP (−3 mmHg) and TPR in combination treatment	Ketosis was the main AE, without DKA GU infections and hypotension were infrequent and manageable
**Heart Failure**							
13	Dapagliflozin vs. Placebo (in addition to GDMT)	DAPA-HF, McMurray et al., 2019 [16]	4744	HFrEF	Multicentre, prospective, randomized, placebo-controlled, double-blind clinical trial, with pre-specified sub-group analysis	Lower risk for primary composite of worsening HF or CV death, 16.3% vs. 21.2%, HR 0.74 (95% CI 0.65–0.85, *p* < 0.001) and secondary endpoints Benefit was consistent across sub-groups, including baseline ARNI or MRA use	No significant differences in volume depletion, renal AEs, or hypoglycemia. Sub-group analysis, including for RAASi drugs was not carried out
14	Empagliflozin vs. Placebo (in addition to GDMT)	EMPEROR-Reduced, Packer et al., 2021 [17]	3730	HFrEF	Multicentre, prospective, randomized, placebo-controlled, double-blind clinical trial, with pre-specified sub-group analysis	Primary composite of CV death or first HF hospitalization was reduced (24.7% to 19.4%) Renal: slower eGFR decline by 1.73 mL/min/1.73 m^2^ per year (*p* < 0.001); renal composite HR 0.50 (0.32–0.77) Benefit additive with ARNI (HR 0.64 ARNI users vs. 0.77 in non-ARNI users: no interaction)	Genital infections more common with empagliflozin No further sub-group AEs reported
15	Empagliflozin vs. Placebo (in addition to GDMT)	EMPEROR-Preserved, Anker et al., 2021 [18]	5988	HFpEF	Multicentre, prospective, randomized, placebo-controlled, double-blind clinical trial, with pre-specified sub-group analysis	Primary composite CV death or first HF hospitalization reduced from 17.1% to 13.8%, HR 0.79 (95% CI 0.69–0.90, *p* < 0.001) Effect was consistent across RAASi use (ACEIs, ARBs, ARNIs, MRAs)	Higher uncomplicated genital/urinary infections and hypotension with empagliflozin Sub-group AE analyses not reported
16	Empagliflozin vs. Placebo (in addition to GDMT)	EMPACT-MI, Hernandez et al., 2024 [19]	6522	Acute MI at risk of HF (new LVEF <45%) or congestion	Multicentre, prospective, randomized, placebo-controlled, double-blind clinical trial, with pre-specified sub-group analysis	No reduction in primary outcome (first HF hospitalization or all-cause death) Reductions in first HF hospitalization HR 0.77 (95% CI 0.60–0.98, *p* = 0.031) and total HF hospitalization rate ratio 0.67 (95% CI 0.51–0.89, *p* = 0.006) Consistent across RAASi sub-groups Reduced need to initiate diuretics, ARNI, ACEI/ARB, and MRA	AEs not reported systematically
17	Ertugliflozin vs. Placebo (in addition to GDMT)	EFFORT, Kang et al., 2024 [20]	128	HF with mildly or moderately reduced EF and chronic functional MR	Multicentre, prospective, randomized, placebo-controlled, double-blind clinical trial, with pre-specified sub-group analysis	Reduced MR severity and improved remodelling indices vs. placebo Effects consistent within ARNI sub-group	Low absolute AE counts No sub-group AE analysis presented
18	Dapagliflozin vs. Placebo (In addition to GDMT), analyzed by baseline HF treatments, including MRA, sacubitril/valsartan, and by ≥50% vs. <50% guideline dose of RAASi, MRA; and by triple/quadruple-therapy combinations	DAPA-HF- therapy-sub-group interaction analysis, Docherty et al., 2020 [21]	4744	HFrEF	Post hoc sub-group analysis of a multicentre, randomized, placebo-controlled, double-blind trial	Primary composite HR range 0.57–0.86 across drug/device sub-groups No treatment-by-sub-group interaction	Volume-depletion AEs slightly higher with loop diuretic or ≥50%dose MRA Renal AEs not increased No other safety interactions
19	Dapagliflozin vs. Placebo (in addition to GDMT), analyzed by baseline sacubitril/valsartan use (ARNI)	DAPA-HF, sacubitril/valsartan (ARNI) sub-analysis, Solomon et al., 2020 [22]	4744	HFrEF	Post hoc sub-group analysis of a multicentre, randomized, placebo-controlled, double-blind trial	Primary composite HR 0.75 (95% CI 0.50–1.13) with ARNI vs. 0.74 (95% CI 0.65–0.86) without; no treatment-by-sub-group interaction Secondary endpoints were consistent irrespective of ARNI	Safety unchanged by ARNI status; no excess volume depletion, renal AEs, hypotension, or hyperkalemia Discontinuation similar SBP, creatinine, and potassium changes were comparable
20	Dapagliflozin vs. Placebo (in addition to GDMT), analyzed by baseline MRA use	DAPA-HF MRA sub-analysis, Shen et al., 2021 [23]	4744	HFrEF	Post hoc sub-group analysis of a multicentre, randomized, placebo-controlled, double-blind trial	Primary composite (CV death or worsening HF) reduced similarly with or without MRA, HR 0.74 in both strata Secondary outcomes were consistent eGFR dip and slope similar (interaction *p* = 0.95); small initial dip	Safety unchanged by MRA use; no excess volume depletion or renal events Moderate/severe hyperkalemia reduced with dapagliflozin in MRA users (1.3% vs. 2.4%, HR 0.50) Mild hyperkalaemia was similar
21	Dapagliflozin vs. placebo (in addition to GDMT), sub-grouped by baseline background; HF therapies counted as 0–1, 2, or 3 classes (ACEI or ARB or ARNI, beta blocker, MRA).	DELIVER, Pabon et al., 2023 [24]	1151	HF with improved EF (HFimpEF)	Post hoc sub-group analysis from a multicentre, randomized, placebo-controlled, double-blind clinical trial	Dapagliflozin reduced worsening HF or CV death across 0–1, 2, or 3 baseline therapy classes Largest relative benefit in 0–1 class sub-group, primary HR 0.46 (95% CI 0.25–0.84). Dosages of existing HF therapies were not reduced due to SGLTi	Adding dapagliflozin did not significantly increase AEs across strata
22	Finerenone vs. Placebo (in addition to GDMT), pre-specified sub-group by baseline SGLT2i use	FINEARTS-HF, Vaduganathan et al., 2025 [25]	6001	Symptomatic HFmrEF or HFpEF	Multicentre, prospective, randomized, placebo-controlled, double-blind clinical trial, with pre-specified sub-group analysis	Finerenone reduced primary outcome (CV death and total worsening HF events), RR 0.84 (95% CI 0.74–0.95, *p* = 0.007) with or without SGLT2i Absolute event reduction nearly doubled with SGLT2i due to higher baseline risk	Hyperkalemia higher with finerenone regardless of SGLT2i Early eGFR dip similar with or without SGLT2i
23	Comprehensive quadruple therapy (ARNI + βblocker + MRA + SGLT2i) vs. Conventional therapy (ACEI/ARB + βblocker)	Combined modelling of EMPHASIS-HF PARADIGM-HF DAPA-HF Vaduganathan et al., 2020 [26]	EMPHASIS-HF: 2, 737 PARADIGM-HF: 8, 399 DAPA-HF: 4744	HFrEF	Cross-trial and actuarial lifetable modelling	Quadruple therapy lowered CV death or first HF hospitalization by 62% vs. conventional therapy, HR 0.38 (95% CI 0.30–0.47) Projected +1.4–6.3 years of life and +2.7–8.3 years free of CV death or HF hospitalization, with larger gains in younger patients	Not reported
24	Baseline ARNI vs. ARNI + SGLT2i (dapagliflozin or empagliflozin)	Fumarulo et al., 2025 [27]	136	HFrEF	Single-centre observational cohort study	ARNI + SGLT2i improved ventricular and atrial volumes, systolic function, and reduced NTproBNP at 3 and 12 months Similar improvements when SGLT2i added to optimized ARNI	Not reported
25	Rapid sequencing of β-blocker + SGLT2i, ARNI, MRA within 4 weeks vs. Conventional sequencing: same four drugs introduced stepwise over ≈ 6 months	NovCon Sequencing Study (protocol), Karamchand et al., 2025 [28]	Anticipated: 584	HFrEF	Single-centre, prospective, randomized, double-blind randomized clinical trial	Results pending on primary: composite of all-cause death + HF hospitalization. Secondary: CV death, 6 min walk, NYHA class, Kansas City Cardiomyopathy Questionnaire (KCCQ) score, echo parameters, NT-proBNP	No reference to systematic evaluation of adverse effects

## 3. Mechanistic Insight

To contextualize the clinical findings, we next summarize mechanistic evidence from animal, cellular, and human studies, showing how RAAS and SGLT2 pathways crosstalk and converge on renal haemodynamics, oxidative, inflammatory, and fibrotic signalling, thereby providing a biological rationale for additive cardiorenal protection with dual therapy.

### 3.1. Effects of SGLT Inhibition on RAAS-Induced Cardiorenal Pathologies

Emerging evidence from experimental animal models and cultured cells indicates that SGLT2 inhibition affords protection in renal and CV pathologies instigated by RAAS activation. In a rat model of chronic Ang II-induced kidney damage, Reyes–Pardo et al. [30] reported that renal SGLT2 mRNA rose in parallel with proteinuria, hyperfiltration, and reactive oxygen species (ROS) formation. Empagliflozin alone did not lower BP and produced only modest benefits in kidney damage; however, when it was combined with the ARB losartan, the protective effects were markedly greater than with either agent alone, supporting additive antioxidant benefits of dual blockade [30]. A similar RAAS–SGLT interaction was documented in porcine coronary artery endothelial cells cultured in high glucose (HG) [31]. Empagliflozin effectively prevented the HG-induced oxidative burst and the pro-senescent phenotype. Perindoprilat or losartan produced similar protection, and this was associated with reduced SGLT1/2 expression, reinforcing the concept that RAAS activity governs SGLT expression and that SGLT-dependent sodium–glucose influx fuels oxidative stress [31]. Extending these mechanistic observations, subsequent studies showed that empagliflozin prevented Ang-II-induced glomerular and tubulo-interstitial fibrosis [32], as well as myocardial hypertrophy and fibrosis [33]. These benefits were associated with reduction in monocyte/macrophage inflammatory infiltrates and [32] decreased sympathetic response [33]. Mechanistically, the benefits of dapagliflozin and empagliflozin are related to reduced activation of the transforming growth factor-β1/SMAD family signalling proteins (TGF-β1/Smad) pathway, which governs extracellular matrix deposition [34], and to lower levels of inflammatory mediators, adhesion molecules, matrix-remodelling enzymes, and RAAS markers, including ACE and Ang I receptor (AT1R) [35]. The TGF-β1/Smad pathway was similarly implicated in dapagliflozin’s favourable effects on Ang II-induced atrial fibrillation, atrial dilatation, and fibrosis [36]; in that model, SGLT2 inhibition also restored electrical activity in the dysfunctional atrium through reduction in the oxidized Ca^2+^/calmodulin-dependent protein kinase II (ox-CaMKII) pathway and normalized ion-channel expression [36]. Additional mechanistic depth came from a more recent study in knock-out mice and human aortic smooth muscle cells, which demonstrated that empagliflozin requires intact fibroblast growth factor (FGF21) signalling to prevent Ang-II-driven vascular pathology. This is a crucial signalling pathway with protective effects in oxidative stress, hypertrophy, vascular wall thickening, and fibrosis through downregulation of TGF-β and other pro-fibrotic proteins (pSMAD2/3, COL1 A1) and upregulation of antioxidant defences (e.g., nuclear factor erythroid 2-related factor 2 (Nrf2), superoxide dismutase 1 (SOD1)) [37]. Consistent with its anti-fibrotic actions, empagliflozin downregulated the Na^+^/H^+^ exchanger (NHE), a key regulator of intracellular pH and cell volume [37], and both empagliflozin and dapagliflozin inhibited fibroblast to myofibroblast transition via sirtuin 6 (SIRT6) upregulation, a histone deacetylase that negatively regulates oxidative stress, fibrosis, and hypertrophy [38]. Finally, dapagliflozin conferred anti-apoptotic effects in Ang-II-treated rats by downregulating store-operated calcium channels (SOCCs), thereby limiting cardiomyocyte apoptosis and adverse remodelling [38,39], suggesting that in clinical use, RAAS blockade, complemented by SGLT2 inhibition, may have additive anti-apoptotic effects. Collectively, these pre-clinical data demonstrate that SGLT2 inhibitors blunt a spectrum of Ang II-driven pathogenic processes, including oxidative stress, sympathetic over-activity, endothelial inflammation, and fibrosis in the renal and CV systems. These benefits arise without lowering BP [30,32,33,34,35,39] or blood glucose levels [30,32,33,34] and are potentiated by concurrent RAAS inhibition [30].

Beyond Ang II models, pre-clinical and clinical data, recently reviewed by Lee et al. [40], show that broader SGLT2-mediated immunomodulation, including suppression of the nucleotide-binding oligomerization domain-Like Receptor 3 (NLRP3) inflammasome (a pro-inflammatory multiprotein complex), reduced levels of interleukin-1β (IL-1β) and Tumour Necrosis Factor-α (TNF-α) and attenuated the canonical transcriptional hubs for inflammatory gene expression, namely nuclear factor kappa B (NF-κB), mitogen-activated protein kinase (MAPK), and Janus kinase/signal transducer and activator of transcription (JAK/STAT) signalling, with reductions in interleukin-6 (IL-6). Antioxidant effects of SGLT2i in cardiorenal disease are also increasingly recognized, implicating broad redox protection via downregulation of nicotinamide adenine dinucleotide phosphate (NADPH) oxidase, as well as the activation of Nrf2/SOD1, catalase, and glutathione peroxidase; suppression of NLRP3-driven oxidative inflammation, attenuation of mitochondrial ROS with improved AMP kinase (AMPK), SIRT signalling, and reductions in lipid peroxidation biomarkers are documented in clinical studies [41]. These pathways may represent additional mechanistic mediators of synergistic effects of dual inhibition.

### 3.2. Effects of SGLT and RAAS Inhibition on Renal Haemodynamics, Fluid, and Electrolyte Balance

Mechanistic studies in humans and animal models suggest that the beneficial effects of SGLT2 inhibition in cardiorenal disease are related to improvement in renal haemodynamics, natriuresis, correction of hyperfiltration, and reduced intraglomerular pressure. In patients with T1D, with hyperfiltration, Cherney et al. showed that empagliflozin reduced GFR by about 33 mL/min/1.73 m^2^ during clamped euglycemia in the hyperfiltering stratum, with parallel decreases in effective renal plasma flow and an increase in renal vascular resistance [42]. Similarly, in mice with unilateral nephrectomy (UNx), empagliflozin lowered hyperfiltration by 36%. However, empagliflozin failed to reduce hyperfiltration or limit albuminuria and fibrosis in UNx/deoxycorticosterone acetate (DOCA) salt models marked by extracellular volume expansion and high salt intake, indicating that hypervolemia and dietary sodium can attenuate SGLT2i-linked renoprotection [43]. The authors postulated that RAAS suppression, in volume-expanded, high-salt states, lowers AT II at the macula densa and afferent arteriole, which reduces the tubuloglomerular feedback gain that SGLT2 inhibition relies on to constrict the afferent arteriole and lower glomerular capillary pressure. They further proposed that hypervolaemia could increase ANP signalling, which can further dampen tubuloglomerular feedback sensitivity [43].

In patients with T2D and HF, ertugliflozin increased 24 h urinary sodium at week 1 versus the placebo (*p* = 0.032) and increased urine volume (*p* = 0.009), which were attenuated at week 12. Yet, SGLT2 inhibition did not raise fractional excretion of lithium (a marker of proximal tubule reabsorption) or total fractional sodium excretion at weeks 1 or 12. By week 12, ertugliflozin reduced extracellular fluid, estimated plasma volume, and supine mean arterial pressure, suggesting a medium-term shift toward euvolemia rather than persistent natriuresis [44]. Evidence suggests that these benefits are complemented by RAAS inhibition. Short-term canagliflozin, when added to background therapy with metformin and RAASi, in patients with T2D produced a measurable contraction in estimated plasma volume at week 1 compared with placebo, which is an effect largely attenuated by week 12. While canagliflozin reduced plasma volume transiently, in agreement with the aforementioned study by Lytvyn et al. [44], glucosuria and fasting glucose effects persisted over the 12 weeks [45]. Similarly, when dapagliflozin was added to background ACEI or ARB in albuminuric T2D, urinary glucose excretion and urinary osmolality increased, while sodium reabsorption at the proximal tubule was inhibited, as indicated by increased fractional lithium excretion. Free water clearance fell, while renin and copeptin, an arginine vasopressin marker, rose by 46.9%, as compared to the placebo group, with a profile consistent with combined osmotic and natriuretic diuresis with physiologic counter regulation [46]. The benefits of dual therapy on renal haemodynamics were also reported in hospitalized patients with decompensated HF and T2D, who were started on SGLT2 inhibitors. eGFR improved over 12 months in 24 of 40 patients, with improvement associated with lower baseline B-type natriuretic peptide (BNP) and concurrent RAAS inhibitor use, suggesting that RAAS background influences renal trajectories on therapy [47]. The haemodynamic advantages of SGLT inhibition (empagliflozin) and RAAS inhibition (ramipril) resulted in decreased cardiorenal injury and reduced inflammatory biomarkers in T1D [48].

Another important aspect of dual therapy, at the level of the kidney, is the effect on afferent/efferent arteriole tone. While RAASi primarily act on the efferent arteriole, SGLT2i reduces afferent arteriolar dilation via tubule-glomerular feedback, lowering single-nephron GFR. These complementary actions reduce intraglomerular pressure and filtration stress. This was confirmed by Wada et al., in Zucker diabetic fatty rats [49]. High-resolution intravital imaging demonstrated that luseogliflozin selectively corrected afferent vasodilation and reduced single-nephron GFR without altering efferent calibre. Adenosine A1 antagonism abrogated these effects, suggesting that the effects of SGLT2 inhibition on afferent arteriolar vascular tone are mediated through adenosine. Co-administration with telmisartan still limited abnormal afferent dilation [49], supporting afferent efferent complementarity under dual therapy. Consistently, in a severe diabetic nephropathy model in Dahl rats, combined SGLT2 inhibition (luseogliflozin) and RAAS blockade (lisinopril) provided greater renoprotection than either agent alone, with attenuation of hyperfiltration and structural injury. Synergistic effects extended to BP lowering, whereby combination therapy reduced BP to a greater extent than lisinopril alone. Luseogliflozin alone had no BP effect [50].

These haemodynamic changes have direct and clinically significant nephrological implications. By lowering intraglomerular pressure via afferent arteriolar feedback, SGLT2 inhibition reduces mechanical stress on the glomerular filtration barrier and decreases albumin and non-albumin protein excretion. In CKD, anti-proteinuric effects are independently associated with improved long-term renal outcomes and a slower rate of disease progression. Recent data in the complex pathophysiological setting of kidney transplantation showed that dapagliflozin reduced urinary albumin and other protein fractions, independent of glycaemic status, further supporting the anti-proteinuric effects of SGLT2 inhibition [51].

These data support a coherent mechanistic framework. SGLT2 inhibition corrects afferent side drivers of hyperfiltration via adenosine A1 and tubuloglomerular feedback, RAAS blockade reduces efferent tone and systemic pressure, and early osmotic diuresis with natriuresis transitions toward euvolemia with neurohormonal counter regulation, with neutral to modestly low effects on plasma volume over time and infrequent volume-depletion events on RAAS-based background therapy [4,10]. The balance of these processes is modulated by salt intake, volume status, and patient-level inflammatory or injury profiles. Limitations include small sample sizes, short treatment windows in mechanistic trials, and species- or model-specific features in pre-clinical work. Nonetheless, the mechanistic complementarity of RAAS blockade and SGLT2 inhibition provides a rational basis for dual therapy to normalize intraglomerular haemodynamics and preserve kidney structure and function.

### 3.3. Effects of SGLT2 Inhibition on Activation of the RAAS Pathway

Mechanistic and clinical studies suggest that SGLT2 inhibition interfaces with both the classical RAAS pathway and the protective ACE2/angiotensin-1−7/Mas axis (Figure 1). The protective arm of the RAAS counterbalances the effects of the classical pathway through vasodilation, natriuresis, and anti-proliferative, anti-oxidative, anti-inflammatory, and anti-fibrotic effects [52]. Published findings on net RAAS activation are mixed. Several studies indicate suppression of the classical pathway in the kidney or heart. In a double-blind, placebo-controlled clinical trial in patients with T2D, dapagliflozin reduced Ang II and angiotensinogen with concurrent falls in BP, while other vasoactive markers were largely unchanged [53]. In T2D animal models, dapagliflozin and canagliflozin reduced Ang II, angiotensinogen, and AT1R expression, and attenuated oxidative stress, inflammation, and fibrosis, which are consistent with findings of suppression of the intrarenal classical axis [54,55]. Other studies point to increases in RAAS indices. In CKD, rises in renin, Ang II, aldosterone, and copeptin were noted without higher 24 h sodium or volume excretion, consistent with compensatory hormonal responses that limit sustained natriuresis [56]. RAAS pathway activation was also reported in normotensive non-diabetic volunteers [57]. Early signals were captured in a six-day study in T2D in which canagliflozin produced transient natriuresis with increased plasma renin activity and reduced natriuretic peptides, suggesting acute volume contraction with counter regulation [58]. Recent mechanistic work further informs the balance between classical and protective axes. In a double-blind trial in patients with T2D and HF, ertugliflozin under clamped euglycemia raised circulating angiotensinogen and ACE and increased urinary ACE2 activity, while by 12 weeks, extracellular fluid and estimated plasma volume fell, indicating concurrent upstream activation and engagement of the protective axis during a shift toward euvolemia [44]. Similarly, in a CV disease mouse model, empagliflozin improved cardiac structure and reduced oxidative and endoplasmic reticulum stress, with a shift from the classical pathway (reduced renin, ACE, and AT1R) toward the protective pathway, with higher AT2R and Mas receptor expression [52]. In a four-week randomized comparison with hydrochlorothiazide in hypertensive T2D, dapagliflozin increased diuresis, glycosuria, and osmolar clearance without altering sodium excretion or glomerular filtration rate; plasma aldosterone increased while plasma renin activity did not change significantly. Dapagliflozin also modulated circulating microRNAs, notably miR30e5p and miR200b, which may have influenced classical and protective RAAS axes, although ACE2 and angiotensin (1–7) were not measured [59].

Changes in RAAS markers have also been reported in patients on SGLT2i therapy. In a randomized trial of hypertensive T2D on ACE inhibitor or ARB therapy for 24 weeks, dapagliflozin increased plasma renin activity relative to control, while aldosterone did not differ between groups, suggesting compensatory activation despite background RAAS inhibition [60]. In a randomized placebo-controlled heart failure trial using liquid chromatography–mass spectroscopy (LCMS) peptide profiling, empagliflozin increased Ang I, Ang II, angiotensin (1–7), and angiotensin (1–5) in patients receiving ARBs, whereas no change occurred on ACEI background, consistent with ARB-mediated renin stimulation providing substrate for ACE2 and shunting toward protective metabolites [61]. Lastly, some studies report minimal net change in composite RAAS indices with SGLT2 inhibition. In patients with T2D and hypertension, the aldosterone-to-renin ratio did not change within two to six months after SGLT2 inhibitor initiation [62]. Consistent with these findings, in a two-week randomized crossover study in patients with T2D and chronic stable HF, empagliflozin increased fractional sodium excretion and reduced blood and plasma volumes, yet there was no evidence of neurohormonal activation of the RAAS pathway, as indicated by stable renin activity, total renin, and aldosterone [63]. The discrepant findings are limited by study design and low number of participants and should be interpreted with caution. The net effect on the classical and protective pathways may depend on timing after initiation, background RAASi therapy, and disease phenotype. Overall, available data support indirect crosstalk between SGLT2 and RAAS through transcriptional regulation, physiological feedback mechanisms, and shared signalling pathways, rather than any direct physical interaction between RAAS proteins and the SGLT2 transporter. The mechanisms contributing to the synergistic effects of RAAS and SGLT2 inhibition in the heart and kidney are summarized in Figure 2.

## 4. Discussion and Future Directions

Current evidence from clinical trials indicates that adding a SGLTi to background therapy that targets RAAS provides additive benefits to cardiorenal outcomes in T2D [6,8], HF [16,17,18], and CKD [9,11], without raising significantly adverse-related effects [10,16]. However, current recommendations are based on post hoc analyses of larger clinical trials or pre-specified sub-group analyses. The few dedicated trials of RAASi plus SGLT2i in diabetes are small and focused on mechanistic or renal surrogate endpoints [14,15], and there is no large factorial trial in HF that isolates the interaction of the two classes. Future work should include adequately powered randomized studies that directly compare RAASi plus SGLT2i with each component alone across HF phenotypes and T2D stages. While all studies, looking at potential synergies with dual inhibition, report clinical outcomes, safety of combination treatment is not well-studied. Safety surveillance should be rigorous and standardized with pre-specified safety analyses for drug-class-anticipated effects, such as hyperkalaemia and volume depletion, and other adverse effects. Understanding the effects on potassium levels, a clinically significant electrolyte, is key to ensuring safe SGLT2i use, in combination with RAAS blockade. Complex physiological processes underlie the effects of SGLT2i on potassium balance, and their use is predicted to simultaneously promote and limit kidney K+ excretion; net clinical effects are neutral to mildly hypokaelemic and protective against hyperkalaemia, especially with RAAS blockade [64]. In DAPA-HF, moderate-to-severe hyperkalemia was reduced among patients receiving steroidal MRAs when an SGLT2 inhibitor was added [23]. SGLT2i delivers more sodium and flow to the distal nephron and maintains euvolemia, so potassium excretion increases despite aldosterone blockade, which explains the lower rates of clinically important hyperkalemia seen when dapagliflozin is layered on top of spironolactone/eplerenone in HFrEF [8]. However, the FINEARTS HF trial [25] reported higher rates of laboratory hyperkalemia with finerenone (non-steroidal MRA) regardless of SGLT2i use [25]. Differences in disease phenotype, patient characteristics, and MRA class (steroidal vs. non-steroidal) may explain these discrepancies; patients in FINEARTS HF, who had HFmrEF or HFpEF, were older, had more CKD and diabetes, and had lower loop diuretic exposure, all of which influence distal K+ secretion. Further studies are needed to understand how combination treatment affects potassium levels and which patient populations may warrant cautious monitoring.

Determining the optimal level of RAAS suppression in dual therapy and evaluating advanced combinations such as triple blockade will be critical for improving treatment strategies and guiding the precise, evidence-based use of combined RAAS/SGLT2 therapy in cardiorenal disease. For example, it is not clear whether SGLT2 inhibition provides differential benefits when combined with either ARNIs, ACEIs, or ARBs, in combination with an MRA. Therapeutic sequencing and timing merit dedicated testing. Pragmatic trials and registry-based randomizations should compare immediate co-initiation of RAASi and SGLT2i, with or without rapid sequencing of all four foundational HF therapies, against conventional stepwise initiation. These designs should incorporate protocolized diuretic management to account for the transition from early osmotic and natriuretic diuresis to medium-term plasma volume reduction, and they should quantify orthostatic symptoms and falls, based on the converging pharmacological effects of these drugs. In the CV system, while studies have focused on HF, promising data in other conditions, such as hypertension [4,5,10], warrant further investigation.

Pre-clinical and human mechanistic data from small studies offer a coherent explanation for these cardiorenal benefits. SGLT2 inhibition restores tubuloglomerular feedback and narrows the afferent arteriole [42,49]; it also reduces oxidative stress [31], inflammation, and pro-fibrotic signalling in the kidney [32,34] and heart [33,36], shifting fluid status toward euvolemia over weeks [44,45]. RAAS inhibition relaxes the efferent arteriole and suppresses classical Ang II signalling These complementary actions lower intraglomerular pressure, limit albuminuria, and reduce cardiac remodelling. While mechanistic studies in animals and cells have provided important insight into the synergistic effects of blocking both pathways, mechanistic endpoints should be embedded in outcome trials to link biological mechanisms to clinical benefit. At the kidney level, studies should quantify changes in tubuloglomerular feedback and afferent–efferent arteriolar tone using validated surrogates, for example, fractional lithium excretion and measured GFR, where feasible, as well as standardized assessments of the initial eGFR dip and chronic slope. Parallel biomarker panels should track and corroborate the intracellular pathways involved (e.g., adenosine A1 signalling, components of the TGFβ1/Smad axis, FGF21, etc.) and explore other potential mechanistic mediators, e.g., NLRP3 inflammasome, IL-1β, TNF-α, NF-κB, MAPK, JAK-STAT, IL-6. In the heart, further investigation of metabolic reprogramming and extracellular matrix remodelling could shed light on how RAAS blockade influences SGLT2i-mediated cardioprotection. It is also important to determine how SGLT2 inhibitors affect intrarenal classical RAAS activity, renin release, and activation of the protective RAAS axis. Through activation of the ACE2/angiotensin-1−7/Mas axis, SGLT2i may promote vasodilation, natriuresis, anti-inflammatory, and anti-fibrotic effects [52,61]. The balance of the classical and protective RAAS axes varies by timing [44,58,61] and can further inform therapeutic regimens that maximize long-term benefits.

To close the evidence gap, an ideal, adequately powered multicentre, double-blind, placebo-controlled randomized trial should enrol adults across HF phenotypes (HFrEF, HFpEF, and HFimpEF) on maximally tolerated GDMT, that includes RAASi, such as sacubitril/valsartan and an MRA. Trial populations should reflect real-world diversity and clinical complexity. Older adults, women, patients with advanced CKD, and those with high salt intake or chronic volume expansion warrant inclusion and stratification because salt load and volume status can blunt renoprotection from SGLT2 inhibition [40]. Patients would be randomized 1:1 to an SGLT2 inhibitor versus matched placebo, stratified by baseline eGFR category, serum potassium category, and loop diuretic dose. The primary clinical endpoint would be the time to the first CV death or HF hospitalization over a median 24–30 months. Key secondary endpoints would include total HF events, Kansas City Cardiomyopathy Questionnaire clinical summary score, arrhythmias, eGFR slope, and a kidney composite that includes a ≥50% decline in eGFR, end-stage kidney disease, or renal death. Safety composites should pre-specify moderate or severe hyperkalemia, symptomatic hypotension, acute kidney injury, and ketoacidosis. Embedded mechanistic sub-studies may include measured GFR, fractional lithium clearance, RAAS peptide profiling by mass spectrometry, and key intracellular inflammatory and pro-fibrotic mediators.

## 5. Conclusions

Dual inhibition of RAAS and SGLT2 is a viable therapeutic strategy for CV and renal protection. Current evidence suggests that combining SGLT2 inhibitors with proven RAAS-blocking medications may improve clinical outcomes in T2D, HF, and CKD by targeting complementary processes that alter haemodynamics, inflammation, and tissue remodelling. Despite the promising available evidence, the field still lacks adequately powered and dedicated randomized trials that isolate the added value of dual therapy versus either component alone. Substantial questions remain about the precise mechanisms of synergy, the optimal therapy combinations, and the patient populations with the highest benefit-to-risk ratio. The next phase should test by design the effects of dual inhibition, integrating mechanistic readouts with hard clinical and safety outcomes in diverse populations in both chronic and acute care settings. Until such data are available, current evidence supports routine use of SGLT2 inhibitors on top of RAAS-blocking therapy (ACEI, ARB or ARNI and MRA), with standard monitoring for kidney function, potassium, and volume status, and with careful adjustment of background diuretic therapy. This strategy is clinically supported and operationally feasible, and it is likely to improve cardiorenal outcomes while definitive trials of deliberate combination therapy are pursued.

## Figures and Tables

**Figure 1 biomedicines-14-00101-f001:**
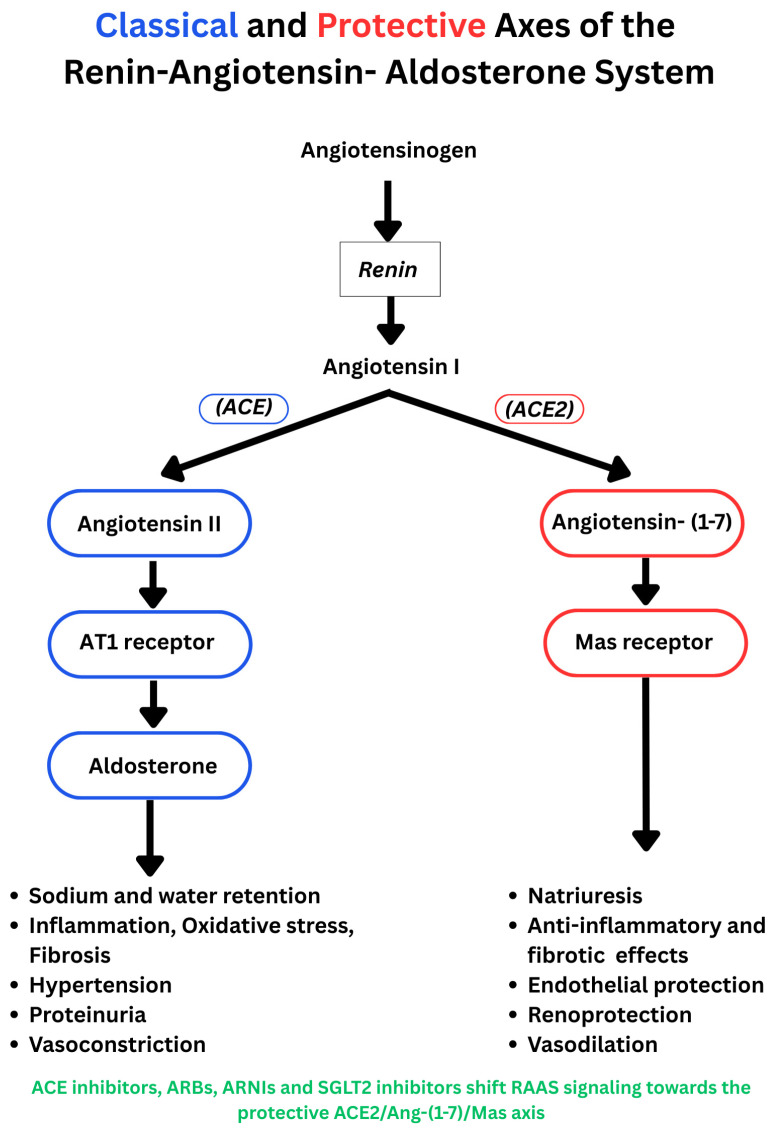
The classical RAAS pathway and the protective ACE2/angiotensin-1−7/Mas axis. (Blue: classical pathway; red: protective axis).

**Figure 2 biomedicines-14-00101-f002:**
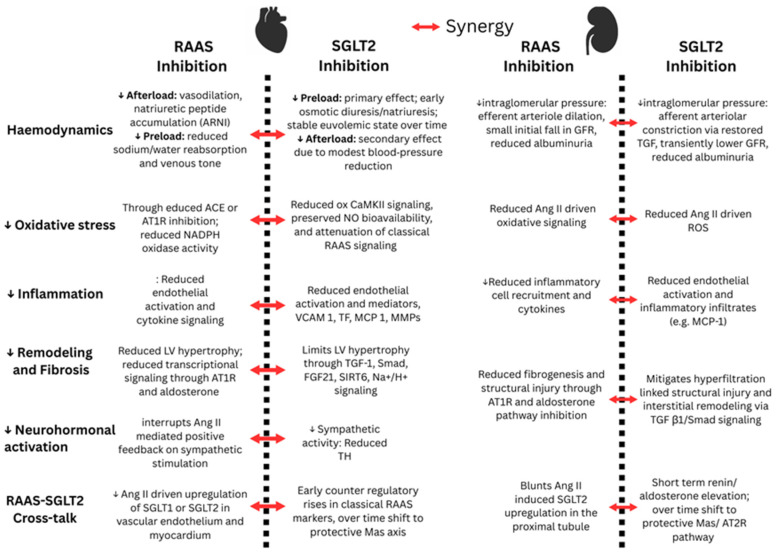
Synergistic effects of RAAS and SGLT2 inhibition in the heart and kidney. The red arrow indicates synergistic effects of dual inhibition (NO: nitric oxide; VCAM 1: vascular cell adhesion molecule 1; TF: tissue factor; MCP 1: monocyte chemoattractant protein 1; MMPs: matrix metalloproteinases; TH: tyrosine hydroxylase).

## Data Availability

Not applicable.

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
