# Peer review of "Dual Inhibition of the Renin–Angiotensin–Aldosterone System and Sodium–Glucose Cotransporter-2: Mechanistic and Clinical Evidence for Cardiorenal Protection"

_biomedicines, 2026, doi:10.3390/biomedicines14010101_

Round 1
Reviewer 1 Report
Comments and Suggestions for Authors
The manuscript provides a comprehensive and timely synthesis of current evidence regarding dual RAAS and SGLT2 inhibition in cardiorenal disease. Its overall organization, progressing from clinical data to mechanistic foundations and concluding with future research priorities, is appropriate for a narrative review. Nevertheless, several areas warrant refinement to enhance clarity, coherence, and adherence to the journal.
The section devoted to clinical trials is extensive but tends to present lengthy, study-by-study summaries that may obscure broader thematic insights. A more selective and integrative approach would strengthen the narrative. Emphasizing overarching patterns, rather than reproducing detailed numerical outcomes for each trial, would improve readability and better serve the objective of a narrative review. Relatedly, introductory or meta-textual statements such as “This section reviews the clinical evidence…” do not contribute substantively and could be removed without loss of content.
The mechanistic overview is thorough and well referenced; however, certain passages provide an extremely granular molecular description that may exceed what is necessary for the intended readership. Streamlining highly specific mechanistic details, particularly when multiple studies converge on similar pathways (e.g., oxidative stress, fibrosis, modulation of inflammatory mediators), would help maintain focus on the most clinically relevant mechanisms while preserving scientific rigor.
From a stylistic perspective, the manuscript would benefit from greater concision. Several concepts recur across sections, for instance, the consistent efficacy of SGLT2 inhibitors across RAAS-inhibitor strata and the absence of major additive safety concerns. While these are important points, their repetition in multiple contexts creates redundancy. Similarly, transitions such as “overall,” “collectively,” and “together” appear frequently and could be reduced to avoid a formulaic tone. A more concise sentence structure would also improve flow.
The reference list is extensive and current, which is a notable strength. Nonetheless, some consolidation may be possible, particularly where multiple preclinical studies address comparable pathways or mechanisms. Including a small number of authoritative systematic reviews or meta-analyses could help contextualize individual findings and balance the evidence base.
A further conceptual point that merits inclusion concerns the nephrological implications of the hemodynamic effects described. While the manuscript clearly illustrates how SGLT2 inhibition restores tubuloglomerular feedback and mitigates glomerular hyperfiltration, it does not explicitly connect these hemodynamic improvements to their most clinically meaningful consequence in nephrology, namely, the reduction in proteinuria. The antiproteinuric effect represents a key therapeutic target in chronic kidney disease, independently associated with long-term renal outcomes, and constitutes one of the most robust benefits observed with SGLT2 inhibitors across diabetic and non-diabetic populations.
Introducing a brief discussion on how correction of intraglomerular hypertension translates into reduced albumin and non-albumin protein excretion would significantly strengthen the mechanistic narrative and better anchor the physiological rationale to clinically relevant endpoints. Recent evidence further reinforces this concept, including data in kidney transplant recipients showing that dapagliflozin lowers urinary albumin and other protein fractions independently of glycemic status. The study by Bilancio et al. (Biomedicines 2025;13:1303) may be a valuable reference to cite, as it highlights the antiproteinuric effect of SGLT2 inhibitors in a population with complex renal physiology and elevated baseline proteinuria.
Incorporating this perspective would broaden the translational relevance of the review and align the mechanistic section more closely with clinical priorities in nephrology.
The discussion appropriately highlights the main limitations of the existing literature, particularly the scarcity of factorial or adequately powered trials specifically designed to interrogate treatment interactions. This section could be further strengthened by articulating more explicit proposals for future research, such as potential trial designs, mechanistic endpoints to be embedded in outcome studies, or subpopulations in which combined therapy should be evaluated. A more integrated synthesis of these considerations at the end of the discussion would yield a clearer set of take-home messages.
In summary, the manuscript is well researched and addresses an important and evolving therapeutic topic. With improved conciseness, reduced redundancy, and a clearer thematic integration the review would be more aligned with academic conventions and the expectations of Biomedicines for narrative review articles.
Author Response
We would like to thank the reviewer for their constructive feedback. We have considered all feedback provided and modified the manuscript accordingly (in tracked changes for ease of reference). We believe that this has improved the manuscript. Our responses are provided point-by-point in bold below.
Comments and Suggestions for Authors
The manuscript provides a comprehensive and timely synthesis of current evidence regarding dual RAAS and SGLT2 inhibition in cardiorenal disease. Its overall organization, progressing from clinical data to mechanistic foundations and concluding with future research priorities, is appropriate for a narrative review. Nevertheless, several areas warrant refinement to enhance clarity, coherence, and adherence to the journal.
The section devoted to clinical trials is extensive but tends to present lengthy, study-by-study summaries that may obscure broader thematic insights. A more selective and integrative approach would strengthen the narrative. Emphasizing overarching patterns, rather than reproducing detailed numerical outcomes for each trial, would improve readability and better serve the objective of a narrative review. Relatedly, introductory or meta-textual statements such as “This section reviews the clinical evidence…” do not contribute substantively and could be removed without loss of content.
We are grateful to the reviewer for raising the important issue of clarity and need for focusing on broader thematic insights and overarching patterns in the clinical trials section. We have carefully reviewed Section 2 (and its subsections: 2.1 and 2.2), in line with the reviewer’s suggestions and the changes are summarized below:
- We deleted non-contributory statements ‘This section reviews the clinical evidence…’ (lines 97-98) and ‘Subsequent cardiovascular studies elucidated … and hypertrophy’ (lines 334-336)
- We synthesized and integrated results on safety across studies (lines 120-124, 136-142, 208-211, lines 256-264)
- We consolidated overlapping content: we merged overlapping findings across CANVAS, EMPA‑REG OUTCOME, and DECLARE–TIMI 58 into a single integrative paragraph, since the studies looked at T2D populations with established symptomatic atherosclerotic cardiovascular disease (ASCVD) or multiple CV risk factors, looking at the common end-point of the composite of CV death and HF hospitalization (lines 129-136). We also synthesized information from the 3 post hoc DAPA-HF studies (lines 255-264). From a stylistic point of view, this has also allowed for concision in the way information is presented.
- We added a sentence for the paragraph starting on line 150 integrating studies focused on kidney and blood‑pressure outcomes in T2D patients with CV disease. EMPA‑REG OUTCOME, and DECLARE–TIMI 58 studies, which were previously in this paragraph, were moved to the aforementioned section under a different theme, as described above.
- We added an introductory sentence (lines 185-186) to aid reader understanding into the next broad theme of results from observational studies.
- We have considered the use of numerical values and deleted non-essential values (lines 112, 223) and consolidated results across studies (line 134). All numerical values are included in Table 1.
- We have revised the text to be more concise throughout (for example 218-232, 240-242, 264-269, 273), in addition to revisions arising from consolidating overlapping efficacy and safety themes, as described above.
The mechanistic overview is thorough and well referenced; however, certain passages provide an extremely granular molecular description that may exceed what is necessary for the intended readership. Streamlining highly specific mechanistic details, particularly when multiple studies converge on similar pathways (e.g., oxidative stress, fibrosis, modulation of inflammatory mediators), would help maintain focus on the most clinically relevant mechanisms while preserving scientific rigor.
We have considered the reviewer’s feedback very carefully and have reviewed Section 3, with an emphasis on revisions to Section 3.1, which described in more detail the mechanistic details in oxidative stress, fibrosis, inflammation. The changes are summarized below:
- We deleted details of the molecular pathways triggered by high glucose, since these are alluded to in the subsequent sentence with less detail (lines 322-325)
- We consolidated, synthesized and categorized the molecular mediators involved in inflammation related to SGLT2i (lines 331-334, 339-342)
- Deleted the specific ion channels modulated by SGLT2i (lines 347)
- Synthesized findings from two different studies on NHE and SIRT6 (lines 355-363)
From a stylistic perspective, the manuscript would benefit from greater concision. Several concepts recur across sections, for instance, the consistent efficacy of SGLT2 inhibitors across RAAS-inhibitor strata and the absence of major additive safety concerns. While these are important points, their repetition in multiple contexts creates redundancy. Similarly, transitions such as (7) general “overall,” “collectively,” and “together” appear frequently and could be reduced to avoid a formulaic tone. A more concise sentence structure would also improve flow.
As described above, we have reviewed the manuscript carefully and made significant changes to improve readability, flow and concision. The specific changes are summarized for both the clinical trials and mechanistic insight sections above. In reviewing the manuscript, we have also reduced redundant/general transitions throughout the manuscript. This has improved readability and flow significantly.
The reference list is extensive and current, which is a notable strength. Nonetheless, some consolidation may be possible, particularly where multiple preclinical studies address comparable pathways or mechanisms. Including a small number of authoritative systematic reviews or meta-analyses could help contextualize individual findings and balance the evidence base.
Thank you for your positive feedback on the reference list and the constructive suggestion to consolidate the list of studies, particularly when it comes to pre-clinical studies. Our review was designed to answer a focused question: what are the mechanistic and clinical effects of deliberately combining SGLT2 inhibition with RAAS targeted therapy in cardiovascular disease. To preserve a transparent link between claims and data, we prioritized primary evidence, excluding review articles for Sections 2 and 3. This is described in the search strategy of the revised manuscript (lines 83-94-specifically on line 89).
Following the helpful comment from the reviewer, we searched for authoritative reviews that align with our research question and scope of this manuscript. We have identified two recent reviews that are closest in scope and describe the molecular pathways where RAAS and SGLT2 inhibitors may cross-talk. These are:
- Forouzanmehr B, Hedayati AH, Gholami E, Hemmati MA, Maleki M, Butler AE, Jamialahmadi T, Kesharwani P, Yaribeygi H, Sahebkar A. Sodium-glucose cotransporter 2 inhibitors and renin-angiotensin-aldosterone system, possible cellular interactions and benefits. Cell Signal. 2024 Oct;122:111335. doi: 10.1016/j.cellsig.2024.111335. Epub 2024 Aug 6. PMID: 39117253.
- Yang X, Qi Y, Hao J, Wei H, Li Z, Xu M, Zhang Y, Liu Y. Effects of oral antidiabetic agents on the renin-angiotensin-aldosterone system. Eur J Clin Pharmacol. 2025 Jun;81(6):801-813. doi: 10.1007/s00228-025-03830-w. Epub 2025 Apr 1. PMID: 40167623.
The first paper usefully catalogs hypothetical cellular links between SGLT2 inhibition and the RAAS, however the focus is on diabetes. The second review surveys several anti-diabetic classes (SGLT2i, GLP‑1RA/DPP‑4i, thiazolidinediones, metformin, sulfonylureas) and their putative effects on RAAS components. Its scope is intentionally broad and diabetes‑centric; it does not focus on deliberate SGLT2+RAAS combination therapy, interaction by background HF therapy, or cardiovascular disease, and it provides limited integration of aspects of interaction (e.g. no focus on TGF-based haemodynamics).
Because our aim is a critical narrative that allows readers to examine critically current evidence, we have retained the full set of primary studies. However, as described above, we have revised the text to consolidate findings and help orient readers to wider thematic areas. We have additionally selectively used review articles, which aim to link the findings of the manuscript to broader SGLT2i literature in regards to immunomodulatory effects (reference 40 in the revised manuscript discussed in lines 373-380), oxidative stress (reference 41; lines 380-387) and potassium balance (reference 64; lines 549-554).
A further conceptual point that merits inclusion concerns the nephrological implications of the hemodynamic effects described. While the manuscript clearly illustrates how SGLT2 inhibition restores tubuloglomerular feedback and mitigates glomerular hyperfiltration, it does not explicitly connect these hemodynamic improvements to their most clinically meaningful consequence in nephrology, namely, the reduction in proteinuria. The antiproteinuric effect represents a key therapeutic target in chronic kidney disease, independently associated with long-term renal outcomes, and constitutes one of the most robust benefits observed with SGLT2 inhibitors across diabetic and non-diabetic populations.
Introducing a brief discussion on how correction of intraglomerular hypertension translates into reduced albumin and non-albumin protein excretion would significantly strengthen the mechanistic narrative and better anchor the physiological rationale to clinically relevant endpoints. Recent evidence further reinforces this concept, including data in kidney transplant recipients showing that dapagliflozin lowers urinary albumin and other protein fractions independently of glycemic status. The study by Bilancio et al. (Biomedicines 2025;13:1303) may be a valuable reference to cite, as it highlights the antiproteinuric effect of SGLT2 inhibitors in a population with complex renal physiology and elevated baseline proteinuria. students
Incorporating this perspective would broaden the translational relevance of the review and align the mechanistic section more closely with clinical priorities in nephrology.
We agree with the reviewer’s observations and have added a paragraph linking the haemodynamic changes to clinically-significant renal outcomes, also integrating the findings from Bilancio et al (Biomedicines 2025;13:1303, reference 51 in the revised manuscript). These are presented on lines 449-457.
The discussion appropriately highlights the main limitations of the existing literature, particularly the scarcity of factorial or adequately powered trials specifically designed to interrogate treatment interactions. This section could be further strengthened by articulating more explicit proposals for future research, such as potential trial designs, mechanistic endpoints to be embedded in outcome studies, or subpopulations in which combined therapy should be evaluated. A more integrated synthesis of these considerations at the end of the discussion would yield a clearer set of take-home messages.
We have now revised the Discussion throughout and consolidated and expanded on explicit, future recommendations at the end of the discussion to close the evidence gap (lines 613-629).
In summary, the manuscript is well researched and addresses an important and evolving therapeutic topic. With improved conciseness, reduced redundancy, and a clearer thematic integration the review would be more aligned with academic conventions and the expectations of Biomedicines for narrative review articles.
Reviewer 2 Report
Comments and Suggestions for Authors
The manuscript entitled "Dual Inhibition of the Renin-Angiotensin-Aldosterone System and Sodium–Glucose Cotransporter-2: Mechanistic and Clinical Evidence for Cardiorenal Protection" in which the authors reviewed the cardiorenal protective mechanisms of both Inhibition of the Renin-Angiotensin-Aldosterone System and Sodium–Glucose Cotransporter-2. They found that current data support layering SGLT2 inhibitors onto RAAS based therapy.
The manuscript is important . However, it suffers from some shortcomings that should be modified.
Shortcomings:
*The authors reported that "combination therapy has not revealed new significant safety liabilities , but also points to the fact that the safety of combination treatment is "not well-studied"." Furthermore, the text highlights a potential inconsistency regarding hyperkalaemia, noting that some studies show reduced hyperkalaemia with MRA + SGLT2i , while the FINEARTS-HF trial showed that hyperkalaemia with finerenone was more frequent "regardless of SGLT2i". Please discuss this discrepancy in the Future Directions or Conclusions sections. Do the authors have a mechanistic hypothesis to reconcile the differing hyperkalaemia observations between the MRA studies (e.g., differences between spironolactone/eplerenone and finerenone, or differences in the patient populations, e.g., HF vs. CKD)? Clarifying this safety signal is essential for clinical practice.
* Also, please add in the Future Directions section the design of the ideal, adequately randomized clinical trial including what patient population, primary endpoint, and duration would be necessary to demonstrate the actual beneficial effect of adding an SGLT2i to maximally tolerated, guideline-optimized RAAS-based therapy (e.g., ARNI + MRA).
* Please add a new figure or a table that demonstrates the classical (ACE/Ang II/AT1R) and protective (ACE2/Ang-(1-7)/Mas) RAAS axes. I think it will enrich your review.
* Please add the abbreviations of used names in full name before abbreviation in the whole review.
Author Response
We would like to thank the reviewer for their constructive feedback. We have considered all feedback provided and modified the manuscript accordingly (in tracked changes for ease of reference). We believe that this has improved the manuscript. Our responses are provided point-by-point in bold below.
Comments and Suggestions for Authors
The manuscript entitled "Dual Inhibition of the Renin-Angiotensin-Aldosterone System and Sodium–Glucose Cotransporter-2: Mechanistic and Clinical Evidence for Cardiorenal Protection" in which the authors reviewed the cardiorenal protective mechanisms of both Inhibition of the Renin-Angiotensin-Aldosterone System and Sodium–Glucose Cotransporter-2. They found that current data support layering SGLT2 inhibitors onto RAAS based therapy.
The manuscript is important. However, it suffers from some shortcomings that should be modified.
Shortcomings:
*The authors reported that "combination therapy has not revealed new significant safety liabilities, but also points to the fact that the safety of combination treatment is "not well-studied"." Furthermore, the text highlights a potential inconsistency regarding hyperkalaemia, noting that some studies show reduced hyperkalaemia with MRA + SGLT2i, while the FINEARTS-HF trial showed that hyperkalaemia with finerenone was more frequent "regardless of SGLT2i". Please discuss this discrepancy in the Future Directions or Conclusions sections. Do the authors have a mechanistic hypothesis to reconcile the differing hyperkalaemia observations between the MRA studies (e.g., differences between spironolactone/eplerenone and finerenone, or differences in the patient populations, e.g., HF vs. CKD)? Clarifying this safety signal is essential for clinical practice.
We are thankful to the reviewer for raising this important point of potassium balance, a key electrolyte, with clinical significance. This has allowed us to include a hypothesis for the discrepant findings across studies (lines 549-566).
* Also, please add in the Future Directions section the design of the ideal, adequately randomized clinical trial including what patient population, primary endpoint, and duration would be necessary to demonstrate the actual beneficial effect of adding an SGLT2i to maximally tolerated, guideline-optimized RAAS-based therapy (e.g., ARNI + MRA).
Following the reviewer’s constructive feedback, we have now revised the Discussion throughout and consolidated and expanded on explicit, future recommendations at the end of the discussion to close the evidence gap (lines 613-629). This is very helpful to provide readers with a clear take-home message for future studies.
* Please add a new figure or a table that demonstrates the classical (ACE/Ang II/AT1R) and protective (ACE2/Ang-(1-7)/Mas) RAAS axes. I think it will enrich your review.
Thank you, we have now added a new figure that illustrates the classical vs the protective RAAS axes, which is very helpful for readers to visualize the balance of the two axes (page 11).
* Please add the abbreviations of used names in full name before abbreviation in the whole review.
We have read through the manuscript carefully and added abbreviations of used names in full before each abbreviation. The abbreviations section has also been updated to reflect changes to the manuscript.
Reviewer 3 Report
Comments and Suggestions for Authors
Congratulations to the authors for their review about one of the most debated topic in current literature; here my comments about it:
The review lacks a clearly defined search strategy, including the databases consulted, search terms, and inclusion or exclusion criteria. Even in a narrative review, providing such details, along with the timeframe of the literature search is still fundamental.
The manuscript presents information in a largely descriptive manner. Although the authors summarize individual studies effectively, they not very often quantify relative treatment effects, such as the degree of eGFR improvement or risk reduction. A clearer distinction between post hoc or subgroup analyses and prospectively designed clinical trials is also needed by the authors.
Authors should consider to include a comparative summary table contrasting RAAS inhibition, SGLT2 inhibition, and their combined use on renal and cardiac. Moreover the discussion could be enriched by expanding on the role of ACE2–Ang(1–7)–Mas axis activation and the modulation of inflammatory cytokines (e.g., IL-6, TNF-α) as potential mechanistic mediators. Indeed authors are encouraged to also include the latest evidences regarding the pleiotropic effects of SGLT2i drugs ( doi: 10.1016/j.hrthm.2025.09.023.)
Author Response
We are grateful to the reviewer for their constructive feedback. We have considered all feedback provided and modified the manuscript accordingly (in tracked changes for ease of reference). We believe that this has improved the manuscript. Our responses are provided point-by-point in bold below.
Comments and Suggestions for Authors
Congratulations to the authors for their review about one of the most debated topic in current literature; here my comments about it:
The review lacks a clearly defined search strategy, including the databases consulted, search terms, and inclusion or exclusion criteria. Even in a narrative review, providing such details, along with the timeframe of the literature search is still fundamental.
We thank the reviewer for raising this important point. The search strategy has been added in the revised manuscript (lines 83-94).
The manuscript presents information in a largely descriptive manner. Although the authors summarize individual studies effectively, they not very often quantify relative treatment effects, such as the degree of eGFR improvement or risk reduction.
We agree with the reviewer that quantifying relative treatment effects is helpful for readers to critically appraise the provided information and extend of effects. These have now been expanded throughout and specifically in lines 119, 134, 165, 168-169, 189-190, 196, 276-277).
A clearer distinction between post hoc or subgroup analyses and prospectively designed clinical trials is also needed by the authors.
We agree that this is key information for transparency and interpretation of findings based on study design. The manuscript has been revised accordingly (lines 127-128, 131, 156-157, 163, 215, 251, 255). Table 1 has also been revised to reflect the study design more precisely.
Authors should consider to include a comparative summary table contrasting RAAS inhibition, SGLT2 inhibition, and their combined use on renal and cardiac. Moreover the discussion could be enriched by expanding on the role of ACE2–Ang(1–7)–Mas axis activation and the modulation of inflammatory cytokines (e.g., IL-6, TNF-α) as potential mechanistic mediators. Indeed authors are encouraged to also include the latest evidences regarding the pleiotropic effects of SGLT2i drugs (doi: 10.1016/j.hrthm.2025.09.023.)
We have carefully considered the reviewer’s helpful comment to include a comparative summary table for the effects on each drug class individually and in combination. The synergies of dual therapy, based on current best evidence, are through indirect crosstalk between SGLT2 and RAAS through transcriptional regulation, physiological feedback mechanisms and shared signalling pathways. This is captured in Figure 2 (page 12) in the manuscript and we have clarified the postulated synergistic effects through converging pathways (lines 527-531) to state this more explicitly.
We agree that further discussion on mechanisms underpinning the beneficial effects of dual therapy is important. We have expanded on the role of the protective RAAS axis (lines 608-610) and included a new figure that illustrates the classical vs the protective RAAS axes, which is very helpful for readers to visualize the balance of the two axes (page 11). We have also expanded further on additional potential immunomodulatory and anti-oxidant mediators (lines 373-387 and lines 601-604). The key anti-arrhythmic effects of SGLT2i are also discussed in the revised manuscript based on the findings of the 2025 meta‑analysis (doi: 10.1016/j.hrthm.2025.09.023) on lines 295-300, reference 29.
Reviewer 4 Report
Comments and Suggestions for Authors
This manuscript is a narrative review of interaction between RAASi and SGLT2i and the effects on cardiovascular and renal system. Review gives comprehensive review of clinical studies as well as some insight in pre-clinical (basic) studies.
Some comments: the reduction in glomerular filtration rate may lead to conservation of body fluids. Please comments on that effect of concomitant RAASi and SGLT2i therapy in humans or other possible studies
2. Please discuss SGLT2i effects on oxidative stress individually (without RAASi th). The paragraph 3.1. there is no convincing evidence that SGLT2 provide any antioxidant protection. Please, check and add a paragraph or two with evidences.
3. lines: 292-294"...This is a crucial signalling pathway with protective effects in oxidative stress, hypertrophy, vascular wall thickening and fibrosis through downregulation of TGF-β and other pro-fibrotic proteins (Nrf2, SOD1 pSMAD2/3, COL1 A1) [36]..." Down-regulation of Nrf2, SOD1 may lead to increased oxidative stress and consequences. Please, comment in the manuscript and support with references. Any molecular/cellular data?
4. line 301"Finally, dapagliflozin was shown to confer anti-apoptotic effects in Ang-II treated rats by downregulating store-operated calcium channels (SOCCs)..." this is model of high ANGII, which is opposite of RAASi- Please comment.
5. line 320- "...hypervolemia and dietary sodium can attenuate SGLT2i linked renoprotection" - please describe mechanisms?
6. Paragraph from line 409- further- it seems that compensatory increase in RAAS (or at least renin-angiotensin II) have beneficial effect on cardiorenal health and is induced by SGLT2. Please provide the mechanism of this effect of SGLT2 on RAAS (molecular or cellular if possible, nor haemodinamically). Any direct evidence on interactions? How would receptor/transporter SGLT2 interact with the RAAS?
Author Response
We would like to thank the reviewer for their constructive feedback. We have considered all feedback provided and modified the manuscript accordingly (in tracked changes for ease of reference). We believe that this has improved the manuscript. Our responses are provided point-by-point in bold below.
Comments and Suggestions for Authors
This manuscript is a narrative review of interaction between RAASi and SGLT2i and the effects on cardiovascular and renal system. Review gives comprehensive review of clinical studies as well as some insight in pre-clinical (basic) studies.
Some comments: the reduction in glomerular filtration rate may lead to conservation of body fluids. Please comments on that effect of concomitant RAASi and SGLT2i therapy in humans or other possible studies
We thank the reviewer for raising this important point. Although both RAAS blockade and SGLT2 inhibition are associated with an early, hemodynamically mediated eGFR dip, this does not lead to fluid conservation. Human studies show an early osmotic diuresis and natriuresis with SGLT2 inhibitors, followed by sustained reductions in extracellular fluid and estimated plasma volume and stabilization at a euvolemic set point over weeks. In clinical trials, volume depletion adverse events remained infrequent and comparable to placebo on contemporary background therapy, including RAAS inhibitors. When used together, the afferent–efferent complementarity that lowers intraglomerular pressure occurs alongside net neutral to modestly lower plasma volume in the medium term, provided routine monitoring of diuretics and volume status is performed. We have revised and expanded on the text as follows: ‘early osmotic diuresis with natriuresis transitions toward euvolemia with neurohormonal counter regulation, with neutral to modestly low effects on plasma volume over time and infrequent volume-depletion events on RAAS‑based background therapy [4, 10]’ on lines 460-463 to clarify the above point in the revised manuscript.
- Please discuss SGLT2i effects on oxidative stress individually (without RAASi). The paragraph 3.1. there is no convincing evidence that SGLT2 provide any antioxidant protection. Please, check and add a paragraph or two with evidences.
We are grateful for the reviewer’s comment, which has allowed us to revise the text to describe the important anti-oxidant effects attributed to SGLT2i. This has been included on lines 380-387 and also tied into the Discussion and Future Directions (lines 601-604)
- lines: 292-294"...This is a crucial signalling pathway with protective effects in oxidative stress, hypertrophy, vascular wall thickening and fibrosis through downregulation of TGF-β and other pro-fibrotic proteins (Nrf2, SOD1 pSMAD2/3, COL1 A1) [36]..." Down-regulation of Nrf2, SOD1 may lead to increased oxidative stress and consequences. Please, comment in the manuscript and support with references. Any molecular/cellular data?
Thank you for identifying this discrepancy. We agree and have corrected the sentence to reflect that Nrf2 and SOD1 are antioxidant defences that are upregulated, not downregulated (lines 353-355).
- line 301"Finally, dapagliflozin was shown to confer anti-apoptotic effects in Ang-II treated rats by downregulating store-operated calcium channels (SOCCs)..." this is model of high ANGII, which is opposite of RAASi- Please comment.
Thank you for your comment, which allows us to clarify that these anti-apoptotic effects were demonstrated under an Ang II-stimulated environment. In clinical use, RAAS inhibition would be expected to reduce the upstream Ang II stimulus, while SGLT2 inhibition could still modulate downstream calcium entry pathways, suggesting potentially complementary, RAAS independent cytoprotective mechanisms. This has been clarified in the revised manuscript (lines 363-367).
- line 320- "...hypervolemia and dietary sodium can attenuate SGLT2i linked renoprotection" - please describe mechanisms?
This is an important point and we have now provided a plausible explanation, as put forward by the authors of the study that showed that hypervolemia and dietary sodium can attenuate SGLT2i-mediated renoprotection (lines 400-405).
- Paragraph from line 409- further- it seems that compensatory increase in RAAS (or at least renin-angiotensin II) have beneficial effect on cardiorenal health and is induced by SGLT2. Please provide the mechanism of this effect of SGLT2 on RAAS (molecular or cellular, if possible, nor haemodynamically). Any direct evidence on interactions? How would receptor/transporter SGLT2 interact with the RAAS?
Considering the clinical benefits and crosstalk across broad molecular pathways, this raises the valid question of the molecular and cellular mechanisms responsible for these interactions. We have clarified in the text that ‘available data support indirect crosstalk between SGLT2 and RAAS through transcriptional regulation, physiological feedback mechanisms and shared signalling pathways, rather than any direct physical interaction between RAAS proteins and the SGLT2 transporter’ on lines 527-530.
Round 2
Reviewer 2 Report
Comments and Suggestions for Authors
The manuscript has been improved prior to the previous revision and acceptable for the publication.
Author Response
We would like to thank the reviewer for their positive feedback on the revised manuscript and prior comments, which helped us improve the manuscript.
Reviewer 3 Report
Comments and Suggestions for Authors
Congratulations to the authors for the revised version of the manuscript
Author Response

(The authors gave the same response as above.)
